# Factors associated with infant mortality in Nigeria: A scoping review

**Loveth Dumebi Nwanze**[ID]*, **Alaa Siuliman, Nuha Ibrahim**

Department of Public Health, School of Medicine, Faculty of Education and Health Sciences, University of Limerick, Limerick, Ireland

* 21151946@studentmail.ul.ie

## Abstract

### Background

Infant mortality persists as a global public health concern, particularly in lower-middle-income countries (LIMCs) such as Nigeria. The risk of an infant dying before one year of age is estimated to be six times higher in Africa than in Europe. Nigeria recorded an infant mortality rate of 72.2 deaths per 1,000 live births in 2020, in contrast to the global estimate of 27.4 per 1,000 live births. Several studies have been undertaken to determine the factors influencing infant mortality.

### Objective

This scoping review sought to identify and summarise the breadth of evidence available on factors associated with infant mortality in Nigeria.

### Methods

This review followed the five-stage principles of Arksey and O'Malley's framework. Four electronic databases were searched with no limit to publication date or study type: Ovid MEDLINE, PubMed, CINAHL Complete, and Web of Science. Selected studies were imported into Endnote software and then exported to Rayyan software where duplicates were removed. Included articles were thematically analysed and synthesised using the socioecological model.

### Results

A total of 8,139 references were compiled and screened. Forty-eight articles were included in the final review. At the individual level, maternal- and child-related factors were revealed to influence infant mortality; socioeconomic and sociocultural factors at the interpersonal level; provision and utilisation of health services, health workforce, hospital resources and access to health services at the organisational level; housing/neighbourhood and environmental factors at the community level; and lastly, governmental factors were found to affect infant mortality at the public policy level.

**Competing interests:** The authors have declared
that no competing interests exist.

## Conclusion

Factors related to the individual, interpersonal, organisational, community and public policy
levels were associated with infant mortality in Nigeria.

## 1. Introduction

An improvement in life expectancy for any country reflects a significant reduction in infant
mortality [1]. The number of infants that die before attaining the age of one year for every 1,000
live births in a year is known as Infant Mortality Rate (IMR) [2]. Infant mortality remains a pub-
lic health issue even though the past two decades have shown significant progress in reducing
infant deaths worldwide, from 65 deaths per 1,000 live births in 1990 to 27.4 infant deaths per
1,000 live births in 2020 [3]. Improvements in global IMR have been credited to enhancements
in women's education, healthcare accessibility, access to proper nutrition, and the direct efforts
of some countries to implement interventions that promote prenatal and post-natal health out-
comes, e.g., child immunisations [4]. However, low- and middle-income countries (LIMCs) still
record very high rates of infant deaths, most especially in the sub-Saharan African (SSA) region,
with an average infant mortality rate (IMR) of 50.2 deaths in 2020 [3]. The risk of an infant
dying before one year is estimated to be six times higher in Africa than in Europe [5]. Out of the
1.9 million infants that died in SSA, Nigeria recorded an estimated 548,116 infant deaths in
2019, representing 28% of total infants that died in SSA in 2019 [6].

As a significant indicator for a country's socio-economic development index, IMR is used
to assess the level of improvement in child health [7]. Goal 3.2 of the Sustainable Development
Goals (SDGs) aims to decrease avoidable deaths of children under five years of age to at least
25 per 1,000 live births by 2030 [8]. However, seven years to the deadline, Nigeria's hope of
achieving this goal seems doubtful, as the rate of decline has been slow, from 124.6 deaths in
1990 to 72.2 deaths per 1,000 live births in 2020 [3]. When compared to other countries in
SSA, for instance, South Africa, Kenya and Ghana, this progress seems inadequate as these
countries are already close to achieving SDG 3.2, with respective IMRs of 28, 31 and 35 deaths
per 1,000 live births [9]. Despite being one of the largest economies in Africa, with an esti-
mated GDP of $448 billion in 2019, the socioeconomic indicators of Nigeria remain low, espe-
cially child mortality [10]. Thus, the slow decline in infant mortality rates in Nigeria raises
questions about the barriers that affect the reduction of infant mortality despite the number of
interventions and policies employed to tackle the challenge.

Studies have traced the high mortality rates of infants to demographic and socio-economic
factors, the most common being poverty/socioeconomic status and poor health service deliv-
ery [11, 12]. Other studies have linked the factors associated with infant mortality to mother
and child characteristics. For example, a conceptual framework developed by Matteson et al.
categorised infant mortality risk factors into personal risk factors such as a mother's sociode-
mographic and medical risk factors and community-level factors such as healthcare facilities,
welfare expenditures, rate of poverty and urbanisation [13]. To estimate the burden of societal
factors on neonatal deaths, Upadhyay et al. adopted the three-delay model in their study. The
findings indicated that a delay in seeking care (level 1 delay) and a delay in receiving quality
treatment at a health facility (level 3 delay) were significant contributors to neonatal mortality
compared to a delay in reaching a health facility (level 2 delay) [14]. A review by Louangpra-
dith et al. [15] suggested three categories of predictive factors for infant mortality, namely: dis-
tal factors (i.e. socioeconomic and demographic factors), intermediate factors such as

maternal, fertility and health service delivery factors and lastly, neonatal factors. Furthermore, by adopting the conceptual model of health capability (CMHC) framework, Bugelli et al. [11] found that IMR was affected mainly by social and environmental conversion factors such as rate of employment, level of education, access to health services and professionals, and the quality of prenatal care receive. Therefore, evidence from the literature suggests that infant mortality rates and birth outcomes are linked to maternal health and well-being [16].

A preliminary search of PubMed and MEDLINE databases revealed no scoping review on the topic. Therefore, the aim of this scoping review was to identify and map out the breadth of evidence on the factors influencing infant mortality in Nigeria. Due to the extensive and complex nature of the factors identified in the literature, the main objective of this review was to thematically synthesize evidence using a socioecological model to report the factors affecting infant death in Nigeria.

## 2. Methodology

A scoping review was chosen as the research methodology for this study because it facilitates the collection of a wide range of sources, helps to identify gaps in knowledge and condenses key findings into conclusions that can guide future research [17]. This review followed the five-stage principles of Arksey and O'Malley's framework, namely: defining the research question, identifying relevant literature, study selection, data extraction, and lastly, collating, summarising, and reporting the results [18]. An unpublished protocol was used to plan and conduct the review. Additionally, the summary and reporting of this review were done using the checklist for the Preferred Reporting Items for Systematic Reviews and Meta-Analyses extension for Scoping Reviews (PRISMA-ScR) [19]. This review did not require ethical approval as the findings were based on existing and publicly available literature.

### 2.1 Defining the research question

The review question is structured within the framework of the Population-Concept-Context (PCC) model, signifying that the primary focus of the study is on infants, with the concept under investigation being infant mortality and the geographical context of interest being Nigeria. Thus, to fulfil the overall objective of this study, this review set out to answer the question: What are the factors associated with infant mortality in Nigeria?

### 2.2 Identifying relevant literature

The search strategy included vital concepts such as infant mortality and Nigeria. Keywords were obtained from the medical subject heading (MeSH) and other references. Following an initial search on PubMed, the search strategy used for this review is presented in Table 1. Four electronic databases—PubMed (National Library of Medicine, Bethesda, Maryland, USA),

**Table 1. Search terms and keywords used.**

| Search | Search terms |
|---|---|
| #1 | Infant OR Newborn OR "new born" OR Baby OR Neonate OR Neonatal OR Postneonatal OR Perinatal OR Fetal OR Foetal OR Postnatal |
| #2 | Mortality OR Mortalities OR Fatality OR Death OR Dying OR Loss |
| #3 | Nigeria OR Nigerian OR "Federal Republic of Nigeria" |
| #4 | #1 AND #2 AND #3 |

# Number

Ovid MEDLINE, CINAHL Complete, and Web of Science (Clarivate Analytics, Philadelphia, PA, USA)—were used to conduct a thorough search of the literature. Additional sources used to collect published articles, grey literature, and reports include Sabinet African Journal (Sabinet African Journals), a South African online journal platform that provides a wide range of African research content and the AfricaBib database (AfricaBib Home) that offers a collection of African social science titles. Table 1 shows the search terms used to search for articles on the databases. The reference lists of included articles were also hand-searched for studies that may have been missed from the online database search. The final search date, including hand searching, was 12[th] July 2022.

## 2.3 Study selection

In this study, a systematic approach was employed to select relevant studies. First, studies were imported into Endnote software [20] and then transferred to Rayyan software [21], to identify and eliminate duplicate entries. In cases of duplication, priority was given to the most recent version of the article, which typically contained the most comprehensive information. Subsequently, two independent reviewers screened the titles and abstracts of the identified studies based on predetermined inclusion and exclusion criteria. These criteria were guided by the population of the study, concept of infant mortality, context, publication date, and language of publication.

The population includes infants aged less than one year, covering the entire spectrum from birth to 365 days. This category encompassed neonates (infants aged less than 28 days), early neonates (between 0 and 6 days of life), perinates (ranging from 28 weeks of gestation to the first 7 days of life), and postneonates (between 28 days and 364 days) [22]. Only studies addressing the topic of infant mortality were considered eligible for inclusion.

The geographic scope of this review was limited to Nigeria. In instances where a study encompassed multiple countries, including Nigeria, we included articles only if it was possible to extract and differentiate information specifically pertaining to Nigeria from the data related to the other countries.

In addition to the above-stated inclusion criteria, no restriction was imposed on publication dates. This flexibility was integral to our objective of comprehensively mapping and summarising the determinants of infant mortality in Nigeria.

To maintain methodological rigour, certain types of publications, such as conference proceedings and case reports, were excluded. Additionally, studies that did not directly investigate the factors influencing infant mortality or explicitly focused on children above one year of age were excluded. Furthermore, studies conducted only in settings outside the Nigerian context and those involving animal subjects were not included in this review. The study exclusively encompassed articles that have been published in English.

## 2.4 Data charting

Data extraction was done independently, and information from eligible articles was charted onto an Excel sheet using Microsoft Office Excel software. The data items used to chart essential details include the article's title, first author, publication year, sample size, geographical region, study design, study objectives etc.

## 2.5 Collating, summarizing and reporting the results

Key themes following the socio-ecological model (SEM) were presented in a table and figure using descriptive and thematic analysis. The five levels of the SEM include the individual level, which describes the traits of the individual; the interpersonal level focuses on the relationship

between the individual and social networks; the organisational level focuses on the characteristics of the formal institution or health system as in the context of this review; the community level emphasises the interactions between the organisation and social institutions within the society; and lastly, the public policy level focuses on laws and policies enacted at either state or national levels [23]. This model was chosen for this study because it explains the intricate interaction between people and social systems and offers some implications for the relationship between a person's health outcomes and their immediate environment [24]. According to the five levels of the SEM, this scoping review aggregates and discusses the factors influencing infant mortality in Nigeria. The findings were divided into themes related to individual, interpersonal, organisational, community, and public policy levels. No individual datasets were used in this study; hence no ethical approval was required.

## 3. Results and analysis

### 3.1 Study selection

8,139 references were identified from four electronic databases (PubMed = 3,206, MEDLINE = 2,466, Web of science = 1,753 and CINAHL = 592) and other sources (two online databases Sabinet = 108, and AfricaBib = 14). After removing 4,115 duplicates, the remaining 4,024 references were retained for title and abstract screening. During the screening process, 86% of the references were independently selected for inclusion by two reviewers, while 14% (n = 563) of the remaining decisions conflicted. Reviewers held meetings to discuss their results and were able to resolve 95% of the conflicts. Where consensus was not reached, a third party was invited to reach an agreement for the remaining conflicts. 3,960 articles were excluded due to lack of relevance by title and abstract. The final stage of the study selection was the full-text screening which involved a critical reading of the eligible articles. A total of 61 articles were eligible for full-text screening. Articles were included if the outcome assessed was infant mortality, including neonatal, perinatal, and postneonatal mortality. In contrast, articles were excluded if infant mortality was not the outcome variable, did not address the research question, results were not specific to Nigeria, and if the full text was inaccessible (i.e., were retracted or unavailable). 45 studies met the inclusion criteria, and 3 articles were added by reviewing the reference lists of included articles. Thus, 48 publications were finally included in the review. The process for selecting the studies is depicted in Fig 1 below.

### 3.2 Characteristics of the included articles

Table 2 lists the characteristics of studies incorporated in the scoping review. Based on study design, 58% of the articles were retrospective studies (n = 28), 8% were prospective studies (n = 4), and only one study adopted a quasi-experimental approach. Most of the studies reported findings for all the six geographical regions in Nigeria (58%; n = 28), 11 articles (23%) focused on the South-Western region, while the remaining 9 articles (19%) reported findings for a combination of regions. According to publication year, out of the 48 articles included, 32 were published between 2017–2022 (66%), 8 articles were published between 2011–2016, 5 articles between 2001–2010 and only 3 papers (6%) were published between the year 1989–1995.

### 3.3 Synthesis of results

The factors associated with infant mortality in Nigeria are described in this review using descriptive and thematic analysis. Themes were developed and categorised according to the levels of the socioecological model: individual, interpersonal, organisational, community and public policy. The themes that emerged from the analysis are presented in Table 3, which also

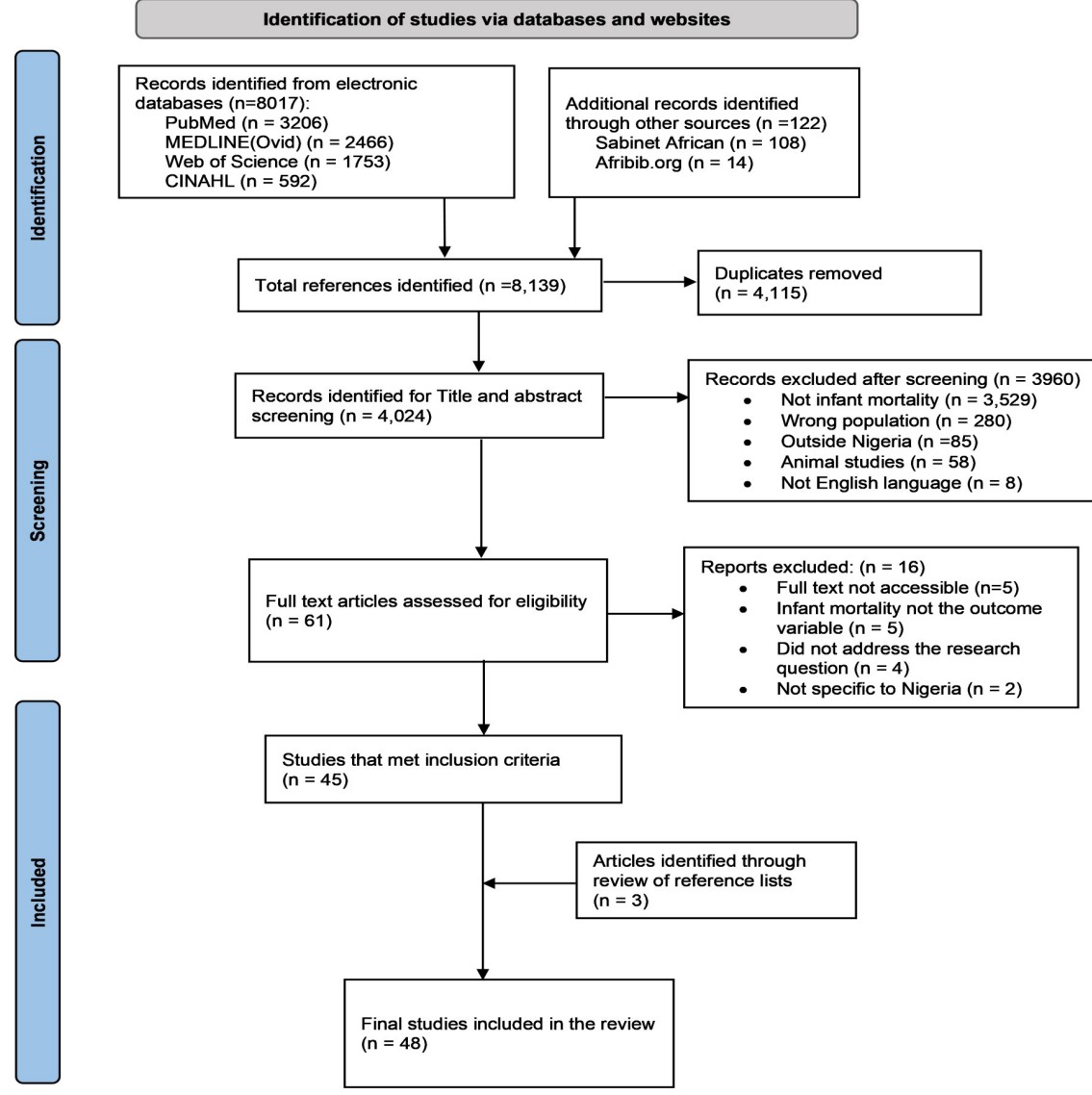

**Fig 1. PRISMA flowchart for the study selection process.**

displays the frequency and percentage of studies for each risk factor identified. The article ID represents the number assigned to each article included in the review. At the individual level, the two themes identified were maternal- and child-related risk factors, while at the interpersonal level, socioeconomic status and socio-cultural factors emerged as themes. At the organisational level, the main themes identified were provision and utilization of health services, access to healthcare services, health workforce, and hospital resources. For community-level factors, the themes identified were housing/neighbourhood factors and environmental factors. Lastly, the risk factors identified at the public policy level were grouped under governmental factors.

**3.3.1 Individual level factors: Maternal- and child-related risk factors.** In this review, thirty-four out of forty-eight articles, found an association between maternal characteristics and infant mortality, while twenty-two articles (approx. 45%) reported on factors related to the

**Table 2. Data Extraction Chart showing studies by title, author/year, sample size, study design, indicator assessed and study objective.**

| Title | First Author | Year | Sample size | Study design | Region | Indicator assessed | Objectives |
|---|---|---|---|---|---|---|---|
| Maternal Employment and Infant Mortality: An Examination of The Role of Breastfeeding as an Intermediate Factor | Bankole A. [25] | 1989 | 2,111 | Retrospective cohort study | South-West | IMR | To assess if breastfeeding is a major intermediate factor in the relationship between maternal employment and infant mortality. |
| Infant Mortality in Nigeria: Effects of Place of Birth, Mother's Education and Region of Residence | Adetunji J. A. [26] | 1994 | 9,727 | Retrospective cohort | North-east, north-west, south-east and south-west | Infant survival | To examine the effects of a child's place of birth, mother's education, region of residence, rural and urban residence on infant mortality in Nigeria |
| Infant Mortality and Mother's Education In Ondo State, Nigeria | Adetunji J. A. [27] | 1995 | 2,635 | Retrospective cohort | South-West | IMR | To discuss the association between maternal education and infant mortality |
| Risk factors for infant mortality in a rural community in Nigeria | Lawoyin T. O. [28] | 2001 | 718 | Prospective community-based study | South-West | IMR, NMR, PNMR | To determine the infant mortality rate (IMR) and factors associated with infant deaths in a rural community. |
| Determinants of neonatal mortality at Wesley Guild Hospital, Ilesa, Nigeria | Onayade A. A. [29] | 2006 | 470 | Comparative study | South-West | NMR | To assess the socio-demographic and other determinants of neonatal mortality |
| A population-based study of effect of multiple births on infant mortality in Nigeria | Uthman O. A. [30] | 2008 | 6,219 | Retrospective cross-sectional study | All six regions* | IMR | To examine if infants of multiple births have disproportionately higher risk of mortality than infants of singleton births. |
| Neonatal Mortality and Perinatal Risk Factors in Rural Southwestern Nigeria | Lawoyin T. O. [31] | 2010 | 972 | Community-based prospective study | South-West | PMR and NMR | To determine neonatal mortality rate and identify perinatal risk factors associated with neonatal deaths. |
| Neonatal Mortality in an Urban Population in Ibadan, Nigeria | Adetola A. O. [32] | 2010 | 1,058 | Prospective community-based study | South-West | NMR | To determine the neonatal mortality rate (NMR), causes of death, and associated risk factors among hospital live births in a suburban population in Nigeria. |
| Determinants of perinatal mortality in Nigeria | Fawole A. O. [33] | 2011 | 9,177 | Retrospective cross-sectional study | North-West, South-West and FCT | PMR | To determine risk factors for perinatal mortality among hospital-based deliveries in Nigeria. |
| Determinants of neonatal mortality in Nigeria: evidence from the 2008 demographic and health survey | Ezeh O. K. [34] | 2014 | 27,147 | Retrospective cross-sectional study | All six regions | Neonatal death | To identify risk factors associated with neonatal death in Nigeria |
| Impact of Socioeconomic Status and Household Structure on Infant Mortality Rate in Abia State of Nigeria | Danawi H. [35] | 2014 | 475 | Cross-sectional Study | South-East | IMR | To measure the impact of socioeconomic status and household structure on infant mortality. |
| Regional Variations in Infant and Child Mortality in Nigeria: A Multilevel Analysis | Adedini S. A. [36] | 2015 | 28,647 | Retrospective cohort study | All six regions | IMR & Child mortality | To examine the effects of individual- and community-level characteristics on child survival during infancy and childhood |
| Risk factors for postneonatal, infant, child and under-5 mortality in Nigeria: a pooled cross-sectional analysis | Ezeh O. K.[37] | 2015 | 66,154 | Retrospective cross-sectional analysis | All six regions | PNMR, IMR, Child mortality | To identify common factors associated with post-neonatal, infant, child and under-5 mortality in Nigeria |
| Trends in neonatal mortality in Nigeria and effects of bio-demographic and maternal characteristics | Akinyemi J. O. [38] | 2015 | 74,060 | Retrospective study | All six regions | NMR | To investigate the trends and factors associated with neonatal mortality in Nigeria |

(*Continued*)

**Table 2.** (Continued)

| Title | First Author | Year | Sample size | Study design | Region | Indicator assessed | Objectives |
|---|---|---|---|---|---|---|---|
| **Delays in healthcare delivery to sick neonates in Enugu South-East Nigeria: an analysis of causes and effects** | Ekwochi U. [39] | 2015 | 376 | Community-based descriptive study. | South-East | Delays in healthcare for newborns | To investigate the factors responsible for delays in healthcare services for sick newborns |
| **Neonatal transport practices in Ibadan, Nigeria** | Abdulraheem M. A. [40] | 2016 | 401 | Prospective and descriptive cross-sectional study | South-West | NMR | To evaluate the modes of transport, pre- and intra-transport care of referred neonates |
| **Determinants of neonatal mortality in rural and urban Nigeria: Evidence from a population-based national survey** | Adewuyi E. O. [41] | 2017 | 30,384 | Retrospective cross-sectional study | All six regions | NMR | To assess the rural–urban differences in neonatal mortality rate (NMR) and the associated risk factors in Nigeria. |
| **Intra-demographic birth risk assessment scheme and infant mortality in Nigeria** | Adebowale A. S. [42] | 2017 | 31,155 | Cross-sectional population-based study | All six regions | IMR | To establish a relationship between Intra-Demographic Birth Risk Assessment Scheme (IDBRAS) and infant mortality in Nigeria. |
| **Neonatal, infant and under-five mortalities in Nigeria: An examination of trends and drivers (2003–2013)** | Morakinyo O. M. [43] | 2017 | 66,158 | Retrospective cross-sectional study | All six regions | NMR, IMR, and Child Mortality | To assess the trends and drivers of NMR, IMR, and U5M over a decade in Nigeria* |
| **Trends and population-attributable risk estimates for predictors of early neonatal mortality in Nigeria, 2003–2013: a cross-sectional analysis** | Ezeh O. K. [44] | 2017 | 63,844 | Retrospective cross-sectional analysis | All six regions | Early NMR | To assess trends in early neonatal mortality (ENM) and population-attributable risk (PAR) estimates for predictors of ENM in Nigeria. |
| **Risk factors for infant mortality in rural and urban Nigeria: evidence from the national household survey** | Adewuyi E. O. [45] | 2017 | 20,449 | cross-sectional analysis | All six regions | IMR | To investigate the rural–urban differences in infant mortality rates (IMRs) and the associated risk factors in Nigeria |
| **Determinants of Infant Mortality in Southeast Nigeria: Results from the Healthy Beginning Initiative, 2013–2014** | Slinkard S. A. [46] | 2018 | 2,436 | Prospective cohort study | South-East | NMR, Early PNMR | Identify determinants for infant mortality among a prospective cohort |
| **Development aid and infant mortality. Micro-level evidence from Nigeria** | Kotsadam A. [47] | 2018 | 294,835 | Quasi-experimental approach | All six regions | IMR | To investigate how Official Development Aid (ODA) affects infant mortality at the subnational level. |
| **Health care at birth and infant mortality: Evidence from nighttime deliveries in Nigeria** | Okeke E. N. [48] | 2018 | 6,867 | Retrospective cohort | All six regions | IMR | To assess the effect of institutional delivery on birth outcomes |
| **Maternal employment and child survival during the era of Sustainable Development Goals** | Akinyemi J. O. [49] | 2018 | 31,828 | Retrospective cross-sectional study | All six regions | IMR and child mortality | To reappraise the relationship between maternal employment and childhood mortality in SSA. |
| **New born care practices and knowledge of risk factors associated with neonatal mortality among postnatal mothers in Ibadan** | Adigun A. S. [50] | 2018 | 206 | descriptive design | South-west | NMR | To assess mothers' knowledge of newborn care practices and factors contributing to neonatal mortality. |
| **Infant mortality and public health expenditure in Nigeria: empirical explanation of the nexus** | David J. [51] | 2018 | NP | Time series | All six regions | IMR | Investigates the nature of relationship between government expenditure on health and the outcome in terms of infant mortality in Nigeria |
| **Child's risk attributes at birth and infant mortality disparities in Nigeria** | Ibrahim E. A. [52] | 2019 | 31,260 | Retrospective cross-sectional study | All six regions | Risk of dying in infancy | To examine the risks of dying in infancy by child's risk attributes at birth and constituent domains |

*(Continued)*

**Table 2.** (Continued)

| Title | First Author | Year | Sample size | Study design | Region | Indicator assessed | Objectives |
|---|---|---|---|---|---|---|---|
| **Community-and proximate-level factors associated with perinatal mortality in Nigeria** | Ezeh O. K. [53] | 2019 | 31,121 | Retrospective cross-sectional study | All six regions | PMR | To identify factors associated with increased PMR. |
| **Effect of oil spills on infant mortality in Nigeria** | Bruederle A. [54] | 2019 | 4,314 | Ecological study | Oil-producing States in the Niger Delta | IMR, NMR and PNMR | To study the causal effects of onshore oil spills before a child's conception on neonatal and infant mortality. |
| **What is the link between malaria prevention in pregnancy and neonatal survival in Nigeria?** | Adeoye I. A. [55] | 2019 | 31,482 | Retrospective cross-sectional study | All six regions | NMR | To investigate the association between malaria prevention in pregnancy and neonatal survival. |
| **Household factors associated with infant and under-five mortality in sub-Saharan Africa countries** | Ekholuenetale M. [56] | 2020 | 31,482 | Retrospective cohort | Nigeria Included | IMR and Child mortality | To examine the influence of household structure on infant and child mortality in SSA. |
| **Impact of mothers' socio-demographic factors and antenatal clinic attendance on neonatal mortality in Nigeria** | Fasina F. [57] | 2020 | 26,826 | Retrospective cohort | All six regions | NMR | To examine the relationship between mothers' socioeconomic and demographic factors on neonatal deaths in Nigeria |
| **Health care expenditure and child mortality in Nigeria** | Adeosun O. T. [58] | 2020 | NP | Time series analysis | All six regions | IMR, NMR | To consider empirically the effects of public health expenditure on the mortality rate of children in Nigeria. |
| **Inequities in child survival in Nigerian communities during the Sustainable Development Goal era: Insights from analysis of 2016/2017 Multiple Indicator Cluster Survey** | Adeyinka D. A. [59] | 2020 | 29,786 | Retrospective cohort | All six regions | NMR, PNMR and Child mortality | To identify the social determinants of age-specific childhood mortalities, which includes neonatal mortality, post-neonatal mortality and child mortality. |
| **Knowledge and practices of immediate newborn care among midwives in selected health care facilities in Ekiti State, Nigeria** | Esan D. T. [60] | 2020 | 89 | Cross-sectional descriptive design | South-West | Newborn care | To assess the knowledge and practices of midwives towards immediate newborn care. |
| **Predictors of infant-survival practices among mothers attending paediatric clinics in Ijebu-Ode, Ogun State, Nigeria** | Sokefun E. E. [61] | 2020 | 386 | Cross-sectional survey design | South-West | Infant survival | To investigate the predictors of infant survival practices among mothers with infants attending paediatric clinics. |
| **Predictors of neonatal mortality in a tertiary institution of a developing country** | Ibor E. K. [62] | 2020 | 3,725 | Retrospective cohort study | South-South | NMR | To determine the causes and factors associated with neonatal mortality. |
| **Public health expenditure and infant mortality rate in Nigeria** | Adeagbo Mathew O. [63] | 2020 | NP | Time series | All six regions | IMR | To examine the trend of infant mortality rate and to estimate the effects of government health expenditure on infant mortality rate in Nigeria |
| **Socio-demographic determinants of post-caesarean neonatal mortality in Nigeria** | Olukade T. [64] | 2020 | 31,828 | Retrospective cross-sectional study | All six regions | CS rates and post-CS neonatal deaths | To estimate the CS rates and the odds of neonatal mortality post-CS delivery in Nigeria; differentiated by selected social determinants of health. |
| **Preventable multiple high-risk birth behaviour and infant survival in Nigeria** | Salawu M. M. [9] | 2021 | 21,350 | Retrospective cross-sectional survey design | All six regions | Infant survival | To investigate the influence of HrBBs on infant survival in Nigeria. |
| **Trends in and determinants of neonatal and infant mortality in Nigeria based on Demographic and Health Survey data** | Patel K. K. [65] | 2021 | 27,465 | Retrospective | All six regions | NMR, IMR | To assess the changes in neonatal and infant mortality rates in Nigeria and examine the socio-demographic determinants |

*(Continued)*

**Table 2.** (Continued)

| Title | First Author | Year | Sample size | Study design | Region | Indicator assessed | Objectives |
|---|---|---|---|---|---|---|---|
| **Trends in the association between educational assortative mating (EAM), infant and child mortality in Nigeria** | Ariyo T. [66] | 2021 | 72,527 | Retrospective study | All six regions | IMR and child mortality | To examine the association between educational assortative mating, infant and child mortality in Nigeria |
| **Effects of environmental crude oil pollution on newborn birth outcomes: A retrospective cohort study** | Eleke C. [67] | 2021 | 338 | Retrospective Cohort Study | South-South | Newborn birth outcomes | To examine the effect of environmental crude oil pollution on newborn birth outcomes |
| **Factors associated with neonatal mortality among newborns admitted in the special care baby unit of a Nigerian hospital** | Ojima W. Z. [68] | 2021 | 1,098 | Retrospective hospital-based survey | South-west | NMR | To evaluate the trends and risk factors associated with neonatal mortality in a teaching hospital in Southwestern Nigeria |
| **Socio-economic determinants of infant mortality rate in Nigeria: Evidence from autoregressive distributed lag technique** | Odjesa E. [69] | 2021 | - | Time series analysis | All six regions | IMR | To examine the socio-economic determinants of IMR in Nigeria in the short and the long-run periods. |
| **Background predictors of time to death in infancy: evidence from a survival analysis of the 2018 Nigeria DHS data** | Kunnuji M. [70] | 2022 | 7,700 | Retrospective cross-sectional study | All six regions | IMR | To explore the predictive nature of bio-demographic risk factors on infant survival in Nigeria |
| **Private health care finance and infant mortality rate in Nigeria** | Cyril O. [71] | 2022 | NP | Analytical/causal research design | All six regions | IMR | To examine private healthcare finance and Infant mortality in Nigeria. |

* All six regions include North Central, North-East, North-West, South-East, South-West, and South-South.

NP Not provided. IMR Infant Mortality Rate. NMR Neonatal Mortality Rate. PNMR Post-Neonatal Mortality Rate. PMR Perinatal Mortality Rate. U-5 Under-five Mortality CS Caesarean section

child. Maternal age at birth was the most reported risk factor among all maternal characteristics identified in the articles, such that 40% of the included articles reported an association between a mother's age at birth and the risk of infant death. Other maternal factors reported in the reviewed articles include maternal BMI, maternal education, fertility rate, health literacy and self-efficacy etc. Among child-related factors, gender/sex of the child, child's weight and immunisation were all factors considered to be strongly related to infant mortality.

**3.3.2 Interpersonal factors: Socioeconomic status and socio-cultural factors.** At the interpersonal level, approximately 38% of included articles reported associations between socioeconomic status and infant mortality, while nine articles (18%) reported on socio-cultural factors. Majority of the reviewed articles that focused on socioeconomic factors at the interpersonal level associated household income/wealth index (n = 14) with infant mortality. Other factors identified include father's occupation, paternal education, maternal occupation/employment, social support (from family) and household headship. Socio-cultural factors identified at the interpersonal level include marital status, religion and ethnicity.

**3.3.3 Organizational level factors: Provision and utilization of health services, health workforce, hospital resources and access to healthcare services.** Twenty-four articles associated provision and utilisation of health services factors with infant mortality, while other articles focused on access to healthcare services and hospital resources. As seen in Table 3, key factors associated with the utilisation of health services include antenatal (ANC) attendance, place of delivery (including traditional birth centres (TBCs) and mission houses) and mode of delivery (c-section vs vaginal). With regards to health workforce factors, the skill of birth attendants was the most reported factor affecting infant mortality in Nigeria. The factors identified under the theme of hospital resources were the adequacy of neonatal transport vehicles and

**Table 3. Thematic analysis of the factors affecting infant mortality in Nigeria based on the SEM.**

| Socioecological Levels | Themes | Risk factors | No of studies | Article ID | % |
|---|---|---|---|---|---|
| **Individual level factors** | **Maternal-related factors** | Maternal age at birth | 19 | 27, 28, 31, 33, 34, 36, 37, 38, 41, 42, 44, 52, 53, 56, 57, 59, 64, 65, 66 | 40 |
| | | Maternal BMI | 2 | 34, 53 | 4 |
| | | Maternal education | 14 | 26, 27, 30, 32, 33, 35, 42, 43, 53, 56, 57, 59, 65,71 | 29 |
| | | Health Literacy & Self-efficacy | 3 | 34, 37, 61 | 6 |
| | | Fertility rate | 1 | 70 | 2 |
| | | Parity | 3 | 42, 59, 65 | 6 |
| | | Gestation age (preterm vs. full-term delivery) | 2 | 40, 46 | 4 |
| | | Preceding birth interval | 14 | 34, 36, 37, 38, 41, 42, 43, 44, 45, 52, 53, 54, 65, 66 | 29 |
| | | Multiple births | 2 | 30, 59 | 4 |
| | | Birth order | 8 | 31, 34, 36, 37, 38, 41, 52, 53 | 19 |
| | | Breast-feeding duration | 2 | 25, 27 | 4 |
| | | Maternal outcome after delivery (died vs. lived) | 1 | 46 | 2 |
| | **Child-related factors** | Child's sex | 13 | 27, 33, 34, 37, 45, 47, 49, 52, 59, 60, 62, 65, 68 | 27 |
| | | Child's weight | 15 | 28, 32, 33, 34, 37, 41, 44, 45, 52, 53, 59, 61, 64, 67, 68 | 31 |
| | | Immunisation | 2 | 51, 72 | 4 |
| **Interpersonal Factors** | **Socio-economic status** | Household income/wealth index | 14 | 26, 28, 30, 31, 33, 36, 38, 40, 43, 47, 48, 59, 65, 68 | 29 |
| | | Father's occupation | 1 | 49 | 2 |
| | | Paternal education | 2 | 37, 41 | 4 |
| | | Maternal occupation/employment | 4 | 25, 29, 41, 49 | 8 |
| | | Social support (Family) | 1 | 61 | 2 |
| | | Household headship | 1 | 56 | 2 |
| | **Socio-cultural factors** | Marital status/marital type | 6 | 42, 45, 49, 50, 53, 56 | 13 |
| | | Religion | 3 | 42, 55, 66 | 6 |
| | | Ethnicity | 2 | 42, 65 | 4 |
| **Organisational factors** | **Provision and Utilisation of Health Services** | ANC attendance | 10 | 29, 32, 33, 41, 46, 49, 50, 57, 65, 66 | 21 |
| | | Lack of prenatal care/proper care at birth | 2 | 26, 33 | 4 |
| | | Place of delivery (TBCs, mission houses) | 7 | 36, 38, 52, 58, 60, 64, 65 | 15 |
| | | Time of birth (Night-time vs. Daytime) | 2 | 54, 60 | 4 |
| | | Mode of delivery (c-section vs vaginal) | 8 | 33, 34, 37, 41, 44, 45, 52, 67 | 17 |
| | | Provision of 24-hr service coverage | 1 | 48 | 2 |
| | | Delay in healthcare delivery | 1 | 39 | 2 |
| | **Access to healthcare service** | Proximity/distance of health facility | 2 | 48, 69 | 4 |
| | | Out-of-pocket spending/Private health finance | 2 | 58, 72 | |
| | **Health workforce** | Availability of obstetricians | 1 | 33 | 2 |
| | | Knowledge level of midwives | 1 | 60 | 2 |
| | | Doctors/patient ratio | 1 | 63 | 2 |
| | | Skill of birth attendants | 7 | 29, 36, 38, 43, 59, 65, 69 | 15 |
| | **Hospital Resources** | Adequacy of neonatal transport vehicles and referral services | 1 | 55 | 2 |

*(Continued)*

**Table 3.** (Continued)

| Socioecological Levels | Themes | Risk factors | No of studies | Article ID | % |
|---|---|---|---|---|---|
| | | Appropriate medical facilities and equipment | 1 | 36 | 2 |
| **Community factors** | **Housing/Neighbourhood factors** | Place of residence (urban vs. rural) | 18 | 26, 35, 36, 37, 38, 41, 42, 43, 44, 45, 49, 55, 56, 57, 59, 64, 65, 71 | 38 |
| | | Region of residence | 9 | 26, 36, 38, 41, 42, 45, 53, 57, 64 | 19 |
| | | Toilet facility | 1 | 43 | 2 |
| | | Access to electricity | 1 | 41 | 2 |
| | **Environmental factors** | Crude oil pollution | 2 | 54, 68 | 4 |
| | | Carbon IV oxide (CO2) emission | 1 | 70 | 2 |
| | | Drinking water source | 2 | 38, 43 | 4 |
| | | Type of cooking fuel | 1 | 43 | 2 |
| | | Conflict within 25km | 1 | 54 | 2 |
| **Public policy factors** | **Governmental factors** | GDP per capita | 3 | 63, 70, 72 | 6 |
| | | Public health expenditure | 4 | 51, 58, 63, 70 | 8 |
| | | External health resources/aid | 2 | 47, 51 | 4 |
| | | **Total no of articles included** | **48** | | |

referral services and the availability of appropriate medical facilities and equipment. Proximity/distance of health facility and out-of-pocket spending/private health finance were the two factors identified under the theme of access to health service.

**3.3.4 Community level: Housing and environmental factors.** Approximately 40% of the reviewed articles reported housing-related factors as a significant community-level factor compared to five articles that reported on environmental factors. Place of residence, which refers to either urban or rural areas of residence, was the most reported factor among the community-level factors examined in the included articles. The studies that reported on environmental factors at the community level linked crude oil pollution, Carbon IV oxide ($CO_2$) emission, drinking water source, type of cooking fuel and conflict within 25km of residence to infant mortality in Nigeria.

**3.3.5 Public policy level: Governmental factors.** At the public policy level, the only theme that emerged was governmental factors. Six articles reported on factors related to the government, with public health expenditure being the most reported factor associated with infant mortality. The other factors identified are GDP per capita and external health resources/aid.

## 4. Discussion

The primary purpose of this review was to identify and describe the breadth of evidence on factors associated with IMR in Nigeria. Findings from 48 articles were charted and synthesised using the socio-ecological model as a framework, as shown in Fig 2. From the review, 11 themes emerged, and 52 categories of factors were identified across five levels of the model. Most articles focused on factors at the individual and organisational levels and very few articles focused on factors at the public policy level. Critical factors identified at socioecological levels and their comparable findings are presented in the following paragraphs.

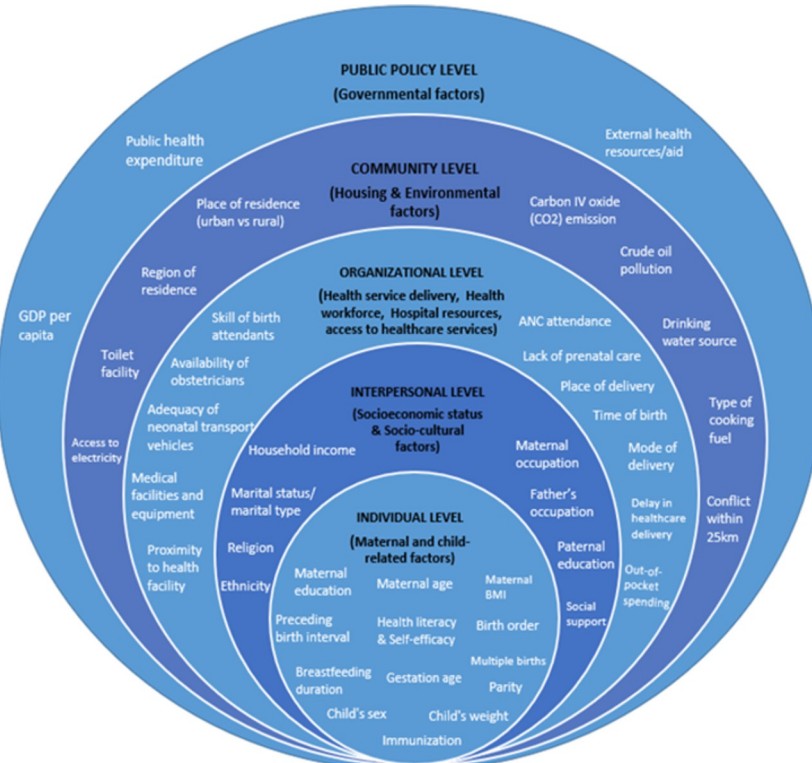

**Fig 2. Adapted socioecological model showing factors associated with infant mortality in Nigeria.**

## 4.1 Individual-level factors

**4.1.1 Maternal-related factors.** *Maternal age and maternal BMI.* Studies in this review reported that mother's proximate factors such as age and BMI were significant risk factors for infant death. Maternal age was observed to have a U-shaped relationship with early neonatal and infant mortality such that infant deaths were more common among younger mothers (< 20 years) and older mothers (> 35 years) [28, 30, 52]. Similar results were reported in Korea [72] Brazil [73] and also in a cross-sectional study that evaluated the association between maternal age and infant mortality across 55 LMICs, indicating that an elevated risk of infant mortality was observed among women below age 18 [74]. These findings can be attributed to the fact that adolescent girls are physically and psychologically immature to handle childbearing and are less likely to make contact with a health clinic [75] while older mothers tend to be more predisposed to hypertension and preeclampsia leading to adverse neonatal outcomes [2, 76]. According to the 2018 DHS report, Nigeria's rate of adolescent pregnancies among girls aged 15 to 19 was approximately 23% [77] which is significantly higher than that of Ethiopia reported to be 13% [78]. Early marriage in Nigeria is synonymous with early childbearing where 4 in 10 women get married before 18 years and 31% have already given birth by age 18 [77].

In the case of BMI, studies in this review reported that babies born to obese mothers (BMI ≥30 kg/m2) had a greater risk of dying than mothers with low BMI [28, 44, 65]. Comparable results were documented in studies conducted in California [2] and Sweden [79]. Maternal obesity has been suggested to lead to higher risks of macrosomia, stillbirth, and neonatal admission to intensive care unit [80].

*Maternal education and health literacy.* From the review, maternal educational status or level of schooling was recognised as a protective factor for infant mortality due to its impact on

the mother's ability to access knowledge that will help promote her baby's health [26, 27, 35, 57]. Similar results were reported in Ethiopia [81], Indonesia [82], and Denmark [83] where better maternal education was associated with a reduction in Infant mortality. In Nigeria, 35% of women (age 15–49) had no education and only about 11% had more than secondary school education in 2018; 88% of the women with more than secondary school education were likely to deliver at a health facility than their counterparts [77]. This suggests that mothers with increased levels of education actively seek health-related knowledge and health system services compared to those with lower or no educational attainment [83]. Education and health literacy help to promote self-efficacy and confidence, allowing adherence to doctors' advice, and increased chances of infant survival [61].

*Parity, fertility rate and gestation age.* Three studies in our review found high parity to be associated with IMR [9, 42, 59], such that babies born to mothers who had many children (grand multiparity) had higher chances of mortality. This is supported by the results of Kozuki et al. where increased mortality risk was observed among high parity infants, especially mothers having high fertility at the end of their reproductive years [76].

Although studies on fertility rate were limited in this study, one study emphasized that in the long run, total fertility rate significantly increased IMR in Nigeria [69]. As of 2018, women in Nigeria gave birth to approximately 5.3 children compared to 6.0 children recorded in 1990 [77]. This suggests that Nigeria's fertility rate has barely changed in the past 33 years. One possible explanation for the high parity and slow decline in fertility rate among Nigerian women is that few women engage in family planning practices. According to the 2018 Nigerian DHS, only about 17% of married women used a family planning method [77], suggesting that the lack of proper family planning could be a potential cause for undesired births (increased fertility rate) and high parity. The likelihood of survival for infants born to high-parity women is severely impacted when there is poor access to adequate care, especially for those with low household wealth [9].

In addition to parity and fertility rate, gestation age which describes the duration of pregnancy before delivery, has been associated with infant mortality. Two studies in the review accentuated that preterm births (< 37 weeks) were a more significant contributor to IMR than term births [32, 46]. A Danish study demonstrated similar results wherein preterm babies were more likely to have an increased risk of death and was commonly observed among mothers with low education, adolescent mothers, and mothers with higher parity among other factors [83].

*Preceding birth interval, multiple births, and birth order.* Fourteen studies in this review demonstrated a significant association between infant mortality and preceding birth interval indicating that women with short birth intervals <18 months had a higher risk of infant deaths. Similar results were recorded in a study that compared birth histories from 66 different countries, revealing that neonatal, infant, and child mortality were strongly related to preceding birth spaces, where births within a short interval after previous birth increased mortality [84]. According to the 2018 NDHS report, 1 in 4 children are born less than two years following the birth of their siblings. Birth intervals of no less than 36 months should be maintained between successive births to lessen the likelihood of infant mortality [77].

Having more than one baby from a single pregnancy, known as multiple births, was reported to be a contributory factor to IMR. Infants from multiple births were considered twice as likely to die compared to singletons [30, 59]. This is comparable to a Korean study that found NMR and IMR to be significantly higher among twin and triplet births than in singleton births [85]. As suggested by Adeyinka et al. one possible explanation for this association is that multiple births tend to lead to preterm births and vulnerability to infection because of low birth weight.

Eight studies highlighted that having a fourth or higher birth order significantly increased the risk of postneonatal deaths [36, 44, 52]. Although similar results were found from research conducted in India, it was suggested that the relationship between birth order and NMR was not linear but J-shaped, signifying that the likelihood of mortality was greater among first and last-born infants compared to those born in the middle [86].

*Breastfeeding duration and maternal outcome.* Breastfeeding was considered by two studies in the review as a protective factor against IMR [25, 27]. Although these studies are old, there is reason to demonstrate the association between IMR and breastfeeding. Findings in the review suggest that a mother's occupation affects IMR through breastfeeding wherein the intensity of breastfeeding matters more than its duration, indicating that the more an infant is breastfed the lower the susceptibility to the risk of death [25]. Another study conducted in Nigeria in 2002 supported these findings, indicating that the higher the level of education and employment of a mother, the higher her chances of breastfeeding her child [87]. Recent research in the U.S. found a significant reduction in postneonatal death across all seven regions in America as a result of breastfeeding [88]. The WHO advocates for exclusive breastfeeding for the initial six months. In Nigeria, 97% of children have been breastfed at some point, with a notable improvement in the practice of exclusive breastfeeding from 17% of infants exclusively breastfed in 2013 to 29% in 2018 [77].

Only one study in the review reported on the association between maternal outcome and infant mortality, emphasizing that mothers who died intrapartum (during childbirth) had an odds ratio of 15.28 when assessing their potential role in predicting infant death [46]. Although it is not clear why studies on this factor were limited, maternal mortality is an important predictor of infant survival and persists as a public health issue. In Nigeria, for every 100,000 live births that occurred in 2018, there were 512 maternal deaths [77], a slight reduction from 576 maternal deaths per 100,000 live births in 2013. A similar study conducted in the Western Pacific Region on maternal and newborn health suggested that providing high-quality care during the period of labour and immediately after childbirth is essential for ensuring the survival of both the mother and the newborn baby [89].

**4.1.2 Child-related factors.** *Child's sex and child's weight.* Child sex emerged as one of the significant predictors of IMR among individual-level factors. Thirteen studies revealed that male infants were more likely to experience mortality compared to female infants. This is consistent with studies from Ethiopia [90] and Angola [12]. Also, recent results from a Brazilian study revealed that male neonates from twin births were at a significantly higher risk of death than females despite weighing heavier at birth [91]. This disparity in IMRs between male and female infants has been ascribed to differences in genetic and biological composition, predisposing boys to disease susceptibility and premature death [53, 90].

Birth weight as a predictor of infant death was highlighted by fifteen articles in the review. Babies with low birth weight (LBW) were highly associated with an increased risk of neonatal death especially for preterm babies. This result is supported by findings from other countries like the U.S. [2], Brazil [92] and Ethiopia [93]. In Nigeria, data on birthweight are not reliable considering that most births occur outside a health facility and data from surveys are based on the mother's assessment of the child's weight [77]. LBW has been attributed to factors like a child's birth order, born to a younger mother, inadequate nutrient intake during pregnancy and poor quality of maternal care [41, 86].

*Immunisation.* Only two studies reported on the protective role of immunisation on infant mortality. These studies indicated that the number of infants immunised against infections and diseases significantly reduced the likelihood of infant mortality [51, 71]. Similarly, a scoping review conducted in Brazil reported the implementation of immunisation programs as a significant determinant of IMR [92]. Even though the rate of basic vaccination has improved

in Nigeria, from 13% in 2003 to 31% in 2018, the number of children who have not received any vaccination has only reduced slightly to 19% in 2018 from 21% in 2013 [77]. Studies have shown that mother's education, literacy level, place of residence, access, and proximity to health facilities were substantive factors associated with immunisation coverage [63, 77, 94].

## 4.2 Interpersonal-level factors

**4.2.1 Socioeconomic status.** *Household wealth index.* Results from fourteen studies demonstrated that households with lower wealth indexes have higher IMRs. Similar trends have been reported by other studies [11, 95, 96]. In Nigeria, household wealth has been shown to influence so many other factors such as education, immunisation, access to antenatal and post-natal care, residence, fertility etc. For instance, when compared to the fertility rate, women from the poorest households give birth to an average of 6.7 children, whereas women from the wealthiest households give birth to approximately 3.8 children [77]. People in poor households are more likely to put off the use of health services as they consider the cost of transportation, medicines, and tests, subsequently leading to a higher risk of mortality for their infants [70].

*Father's occupation and paternal education.* Studies on the relationship between IMR and a father's occupation and education were quite limited in this review. These factors were also reported in a study done in Pakistan to understand the paternal factors associated with neonatal deaths [97]. Akinyemi et al. stressed that the type of father's occupation significantly influenced infant mortality such that infants born to fathers whose occupation was agriculture and labour had a higher likelihood of dying during infancy than fathers who were professionals [49]. One possible explanation for the association between father's occupation and IMR is that fathers who are exposed to certain types of chemicals and pesticides are more likely to have babies with adverse birth outcomes [97].

Regarding paternal education, children whose fathers had no formal education were more likely to die compared to those born to fathers with formal education [53]. This finding is contrary to an earlier study that found no significance in the role of a mother's partner's education and infant mortality [96]. However, in terms of teenage pregnancy, one study found that adolescents whose fathers had no formal education had a higher chance of early childbearing compared to those whose fathers received an education [81]. These results underscore the significant role that a father's level of education plays in mitigating early child marriages, adolescent pregnancies and maternal deaths, which, in turn, can reduce infant mortality rates (IMR).

*Maternal occupation/employment.* Only four studies examined the impact of mothers' occupation on infant mortality. It was revealed that working mothers were more likely than non-working mothers to experience infant deaths [29, 49]. This result contrasts with a study from Japan that showed no link between the likelihood of infant death and the mother's employment, rather, concluded that mothers' occupation impacts infant health outcomes only when they are pregnant [98]. Alternatively, the findings from our review that suggest that maternal employment negatively impacts child survival could be interpreted with reference to the type of occupation. Mothers with low-status occupations like farming and petty trading may experience a high risk of infant death due to low income and unfavourable work conditions that promote poor health outcomes for mother and child compared to mothers in professional and administrative roles [99].

*Social support and household headship.* Support from family was shown by only one study in the review to be a crucial factor in the well-being and survival of infants. Social support from family members was defined as the frequency of receiving assistance and encouragement from family members to comply with health counsel for infant survival [61]. This is supported by studies that associated social support during pregnancy with birth size [100, 101]. It is

suggested that psychosocial distress and depression during pregnancy could cause the production of higher levels of cortisol that could lead to low birth weight and preterm birth. Therefore, social support was found to be a protective factor that acts as a buffer from stress hormones thereby promoting healthy embryonic growth and infant outcomes [102].

Regarding household headship and structure, one study found female household headship to be associated with a reduction in the risk of infant mortality [56]. This is consistent with findings from a study conducted in India where children from female-headed households were found to have a greater chance of survival than children from male household headship regardless of place of residence and child's sex [103]. In Nigeria, the proportion of female-headed households has remained practically the same since 2013 with less than 20% of households headed by women and only about 34% of married women being able to make decisions about their healthcare, paying a visit to relatives and making purchases [77]. It is therefore imperative to acknowledge the role women play in the family and encourage the enhancement of a woman's autonomy and empowerment to enable her to make better decisions regarding utilising health services for both her and the baby, thus, contributing to the reduction of infant deaths in Nigeria [56].

**4.2.2 Sociocultural factors.** *Marital status/marital type*. Results from this review showed that maternal marital status was strongly associated with infant mortality, which is consistent with other studies [104, 105]. Infants born to mothers who were married or in a cohabiting relationship displayed a decreased mortality risk in comparison to those whose mothers were previously married (divorced, widowed, or separated) [45]. Also, children born into polygynous marriages (married to more than one wife) were discovered to have a higher risk of infant mortality [56]. This finding is supported by Smith-Greenaway and Trinitapoli (2014), who confirmed the survival disadvantage of infants born to polygynous marriages irrespective of a higher or lower household wealth index [106]. In Nigeria, the effect of marital type on IMR has been observed to co-exist with the region of residence; in 2013, 36% of women in the northwest were in polygynous unions compared to only 9% of women in the southeast [107]. Polygyny is linked to high parity and high fertility and has been found to impact infant survival through resource limitations and poor living conditions [56].

*Religion and ethnicity*. Religion and ethnicity were identified as predictors of infant mortality. Similar findings were reported in a study in India where infant mortality was strongly associated with religion and caste [86]. Results from this review suggest that children from households that practice Islam had a higher rate of mortality [55, 65] compared to other religions, and children born to the Hausa/Fulani ethnic group were more exposed to death during infancy [9, 42]. This could be attributed to the existence of polygamous marriages, teenage marriages, early childbearing, and high-risk birth behaviours (such as short birth intervals and high parity) among mothers from the Hausa tribe [9].

## 4.3 Organizational-level factors

**4.3.1 Provision and utilisation of health services.** *ANC attendance and prenatal care.* Findings from this review demonstrated the association between increased infant mortality and lack of ANC attendance [29, 33, 46, 50], which is also similar to findings from a Kenyan study where a greater risk of death was observed among neonates delivered to mothers who did not utilise ANC services during pregnancy [108]. Health facilities provide ANC services as a way of monitoring and supervising pregnancies and utilising this service is seen as a protective factor that offers the opportunity to avert, detect, and possibly treat illnesses or problems that increase the likelihood of infant death [32]. In Nigeria, the percentage of women who attend four or more ANC visits has improved, from 44.1% in 2008 to 57% in 2018 [77],

however, the majority of these women were from the urban areas while 47% of rural women were found not to have received any antenatal care. Female education and wealth index are precursory factors that influence ANC attendance [107], therefore, the quality of prenatal care and intrapartum supervision offered during ANC visits proves to be essential factors in mitigating the risks of stillbirths and other adverse birth outcomes [33].

*Place of delivery, mode of delivery and time of birth.* Correspondingly, delivery conditions such as birthplace, time of birth, and mode of delivery were found to influence infant mortality [31, 53]. Mothers who delivered their babies outside the hospital (at traditional birth centres and mission houses), were reported to experience more infant deaths than mothers whose delivery occurred at a health facility [26, 69]. This result is corroborated by a study done in Bangladesh [109]. In Nigeria, the percentage of health facility deliveries has barely improved, where 35% of births were recorded in 2008 compared to 39% in 2018 and over 59% of births are still delivered at home. Place of delivery is largely driven by the level of female education attained and wealth index [77]. One study in the review examined the connection between place of delivery and social class and revealed that home births and deliveries in unsupervised settings were more frequent among women from lower socioeconomic groups than others [68]. This finding demonstrates that mothers from poor backgrounds would seek cheap and substandard healthcare services rather than pay for quality services due to financial limitations [71].

Neonates delivered by caesarean section have a significantly higher risk of death than those delivered normally [38]. This finding is in line with a study done in California [2]. However, it has been suggested that having a skilled professional or birth attendant during delivery at a health facility, especially for neonates born through caesarean section, helps to avert or resolve birth complications [53, 64].

About the time of birth, studies in the review emphasised that women who delivered during the daytime at a health facility, as opposed to nighttime (between the hours of 8 p.m. and 8 a. m.), had a reduced risk of infant mortality [31, 48]. In contrast, a study conducted in Spain found no distinction in terms of survival between births that occurred during the day and those that took place at night [110]. The only plausible reason attributed to the negative relationship between nighttime delivery and IMR was that the lack of proper/facility care at the time of birth increases the risk of neonatal deaths [48].

*Provision of 24-hour service coverage and delay in healthcare delivery.* An increase in the rate of mortality was predominantly observed in areas that lacked access to 24-hour health facilities [48]. Studies in high-income countries have demonstrated that having facilities with medical personnel available 24/7 is a key factor in reducing infant mortality [110, 111]. The importance of providing round-the-clock services could be linked to birth timing, such that women who go into labour at night are given the same attention as those for daytime delivery.

Only one article reported on the delays in healthcare delivery and NMR, however, the findings revealed that delay in reaching a health facility (level-2 delay) was the most significant factor in NMR compared to the delay in receiving healthcare (level-3 delay) [39]. This is somewhat different from the study led by Upadhyay et al. where level-3 delay was shown to be the most important contributory factor to NMR [14]. In Nigeria, delay in healthcare services is mostly related to lack of transportation to and from the facility whereas delay in healthcare delivery was commonly observed among tertiary and secondary health institutions than primary and private hospitals [39].

**4.3.2 Access to healthcare services.** *Proximity to health facilities and out-of-pocket spending.* The influence of household proximity to health facilities was investigated by two studies as a potential factor associated with NMR. Neonates born to mothers residing close to a hospital exhibited a reduction in the likelihood of neonatal mortality compared to neonates born to mothers residing far away [68]. A similar study has also reported accessibility to health

facilities as a determinant of IMR [92]. In Nigeria, access to health facilities was reported by 52% of women as a major concern in 2018, particularly women from rural areas (60%), while 26% of women reported concerns about the distance of the health facility [77].

In the review, studies showed that out-of-pocket spending and dependence on private health finance are associated with IMR and are a major deterrent to seeking healthcare services [58, 71]. This contrasts with another study that found no significant association between private health expenditure and neonatal or infant mortality [112]. Over 97.4% of Nigerian women reported not having any health insurance, relative to 1.9% of women with employer-based health insurance [113]. This suggests that many Nigerians pay for health services out of their pocket and thus calls for the re-evaluation of the National Health Insurance Scheme (NHIS) in Nigeria.

**4.3.3 Health workforce.** *Availability of obstetricians and doctors/patient ratio*. Fawole et al. highlighted the positive effect of the availability of obstetricians on the reduction of perinatal death [33]. The presence of an obstetrician is essential for providing quality obstetric care for both mother and child. In addition, findings from the review showed that an increase in the number of doctors per 1,000 patients caused a reduction in IMR [63]. These findings are corroborated by a Vietnamese study that confirmed the importance of the availability and density of doctors in improving health outcomes [114]. In 2019, Nigeria's doctor per 10,000 population ratio was 3.67, compared to other African countries like Libya (21.57) and Algeria (17.32) [115]. This suggests that the number of doctors available to treat patients in Nigeria is significantly low and contributes to limited access to obstetric and postnatal care, consequently leading to increased mortality rates [63].

*Skill of birth attendants and knowledge level of midwives*. Studies revealed that NMR and IMR mortalities were highest among women whose prenatal care was provided by an unskilled attendant [38, 43]. A similar study demonstrated that having auxiliary nurses as birth attendants compared to doctors was a predictor of IMR [15]. The utilization of skilled birth assistance in Nigeria has slightly improved to 43% in 2018 from 32% in 1990 [77]. Furthermore, knowledge and understanding of midwives on evidence-based newborn care practices play a pivotal role in the reduction of neonatal mortality rates, particularly in Nigeria, where a significant proportion of childbirths (32%) are attended to by a nurse/midwife [77]. It is therefore imperative to empower and upskill birth attendants on best practices offered during childbirth and immediate newborn care, as they serve as essential support in preventing complications and ensuring survival during the birthing process [60].

**4.3.4 Hospital resources.** *Adequacy of neonatal transport vehicles and appropriate medical facilities and equipment*. Results from this review suggest that inadequate neonatal transport services are barely considered to influence infant mortality in Nigeria as seen in the dearth of studies on the topic [40]. A study conducted in India supports this factor, where a substantial reduction was observed in neonatal and infant mortality due to the provision of emergency neonatal transport services [116]. NMR appears to be closely associated with the quality of care provided during pregnancy and delivery, thus, having adequate transport and referral services for neonates is critical, especially for transferring high-risk infants to higher-level facilities [39].

Likewise, the absence of essential medical equipment, such as incubators for the management of low-birth-weight infants, and inadequate power supply to sustain the limited incubators are all contributory factors to IMR [62]. This finding is also supported by a study in Bangladesh [109]. Since comprehensive data on neonatal transport services in Nigeria is limited, the provision of neonatal emergency transport services and medical equipment should be considered paramount for the reduction of IMR.

### 4.4 Community-level factors

**4.4.1 Housing/neighbourhood factors.** *Place of residence and region of residence.* At the community level, place of residence and geopolitical region were found to be associated with IMR, with most of the studies reporting on both factors concurrently. These results are in line with other studies that have reported association between place and region of residence and infant mortality [12, 90, 117]. Studies that assessed the association between place of residence found inequalities in NMR and IMR across all the populations examined. Infants born to mothers residing in rural areas reportedly experienced higher IMRs than mothers who lived in urban areas [26, 35, 44, 57]. In Nigeria, urban areas consistently exhibit lower mortality rates in comparison to their rural counterparts as captured in the Nigeria DHS report. In 2013, IMR in urban areas was 60 deaths per 1,000 live births compared to 86 deaths per 1,000 live births in rural areas [107], while, in 2018, there was a slight reduction in IMR for urban areas (56 deaths per 1,000 live births) and rural areas (74 deaths per 1,000 live births) [77]. Evidence suggests that the reason for this urban-rural disparity in IMRs could be attributed to limited access to healthcare services and lower socioeconomic status among the inhabitants of rural areas [90, 92].

IMRs were observed to be different across the six geopolitical regions. For instance, some studies reported increased IMR in the North-West region compared to other regions [42, 57], while others reported a combination of two or more regions associated with increased risk, such as North-West and North-East regions [45, 64]. A plausible explanation for the variations in IMRs across different regions as reported by the included articles could be accredited to sampling sizes, the region assessed, and the possibility of under-reporting from data sources used. However, it was noted that mothers who reside in northern Nigeria face a higher risk of infant mortality than those who live in the south. According to the 2018 NDHS report, the northwest had an IMR of 80 deaths per 1,000 live births compared to 43 deaths per 1,000 live births in the southwest [77]. This has been attributed to differences in Nigeria's regional environment regarding socioeconomic levels, health-seeking behaviours, and cultural customs [36]. For instance, the northern region of Nigeria is predominantly faced with child marriages, teenage mothers, insurgency, low levels of education and Islamic populations [118], while the southern regions are well known for having higher levels of education, utilising health services, having better socioeconomic development and being Christians.

*Toilet facility and access to electricity.* Access to proper sanitation facilities has been shown to influence infant mortality [43], which is comparable to the results from a multi-country level study [119]. In Nigeria, 56% of households had improved sanitation facilities in 2018 compared to 25% of households with no sanitation facilities [77]. Inadequate sanitation can contribute to the spread of infection which can lead to an increase in the risk of infant deaths [43].

Access to electricity has been shown in this review to play a crucial role in reducing IMR [41]. Similarly, in Ghana, it was observed that a 10% improvement in electricity accessibility reduced IMR, particularly within regions characterized by high mortality rates [120]. In 2018, 59% of households in Nigeria had access to electricity, urban households had more access (83%) than rural households (39%) [77]. Electricity is very important in healthcare as it contributes to the provision of quality care in health facilities through the optimization of medical facilities and equipment [41].

**4.4.2 Environmental factors.** *Crude oil pollution and $CO_2$ emissions.* In the review, three studies demonstrated the association between environmental pollution and IMR such that mothers who resided in areas exposed to crude oil spills and $CO_2$ emissions were found to be at a higher risk of experiencing preterm birth and NMR [54, 67, 69]. A study conducted in Ecuador reported that women who lived in communities exposed to oil pollutants faced a heightened risk of spontaneous abortion [121]. Even though there is insufficient data on crude

oil pollution and mortality rates in Nigeria, most oil-producing regions, specifically the south-south region, often suffer from environmental pollution due to oil exploration and oil spills and evidence suggests that exposure to hydrocarbons poses a significant risk to neonatal development and other adverse health outcomes [54].

Similarly, $CO_2$ emissions from cement production and gas flaring activities were revealed to significantly increase neonatal and infant deaths in Nigeria [69]. Similar studies have also identified a positive correlation between exposure to carbonaceous pollutants and increased infant mortality [122, 123] this is attributed to the fact that carbonaceous emissions have the potential to induce inflammation and oxidative stress that can impact the gestation period and foetal growth during pregnancy [123].

*Drinking water source and type of cooking fuel.* The importance of drinking water sources was emphasized in two studies [38, 43] such that access to an improved source of drinking water in the household was associated with survival advantage among infants. Other studies have noted similar findings on the effect of improved water sources on mothers and infants [93, 124]. As reported in the Multiple Indicator Cluster Survey (MICS) in 2021, approximately 79% of Nigerian households enjoyed access to an enhanced drinking water source, while 15% of households relied on unimproved water sources, reflecting a noteworthy decline from the previous rate of 25% recorded in 2018 [113].

Additionally, only one study demonstrated that women who used biomass/unclean fuel experienced higher neonatal and infant deaths than those using clean fuel [43]. In Nigeria, the percentage of households using solid fuel for cooking (including charcoal, wood, coal/lignite, straw/shrubs/grass, and animal dung) was approximately 65% compared to 27% of households using clean fuel for cooking (including electricity, natural gas, and biogas) [113]. This suggests that more families in Nigeria are exposed to high levels of indoor fumes and air pollution which lead to respiratory problems and other health implications for infants and mothers. Poverty and low wealth index appear to be the main factors influencing the use of solid/unclean fuels [43].

## 4.5 Public policy level

**4.5.1 Governmental factors.** *Public health expenditure and GDP per capita.* The main governmental factor reported in the review was public health expenditure. Similar to studies done in India [125] and sub-Saharan Africa [113], this review found a negative relationship between public health expenditure and infant mortality, such that a percentage increase in government health spending led to a decrease in Nigeria's infant mortality rate [51, 69]. Although the recommended healthcare budget by the WHO in 2009 was pegged at 15% of a country's annual budget, Nigeria's allocation to the health sector dropped from 8.2% in 2014 to 3.9% in 2018 [63]. The effect of this inadequate budgetary allocation to the health sector explains why the Nigerian healthcare system has been in a dysfunctional state, particularly the inadequate provision of health services, the shortage of medical equipment and facilities, wide gaps in supply and demand of skilled health workers, and the over-dependence of the masses on out-of-pocket payments for health services [126].

Likewise, studies on GDP per capita [63, 69, 71] suggest that lower GDP per capita negatively affects access to quality health services and leads to people depending on inadequate, cheap and traditional health services [63]. In 2020, Nigeria recorded a GDP per capita of $2,074.6, a significant decline from $3,201 in 2014. Countries with similar and even lower GDP per capita have recorded lower IMRs compared to Nigeria. For instance, in 2020, Bangladesh with a GDP per capita of $2,233.3 had an IMR of 24.06 deaths per 1,000 live births while India recorded a GDP per capita of $1,913.2 and an IMR of 26.82 deaths per 1,000 live births

[3, 127]. This disparity in IMRs demonstrates the complexity between economic development and basic infrastructure and therefore suggests that Nigeria's GDP has not translated into an improvement in healthcare provision and infant survival.

*External health resources/aid.* This review found the effect of external health resources to influence infant mortality directly and indirectly [51]. Comparably, results from a recent study revealed that health aid showed a favourable and significant impact on the health outcomes of low-income countries [128]. In Nigeria, external health aid from donor agencies and non-governmental organisations was reported to significantly affect infant mortality when directly invested in the health care system. Likewise, external aid in the form of aid projects such as agricultural, energy generation, water, sanitation, and health projects was shown to have an indirect effect on IMR [51]. The likelihood of death is lower for children born in areas close to an ongoing aid project than those without an aid project. Therefore, the allocation and effectiveness of external health resources in Nigeria tend to vary across different regions, and the impact of such aid on IMR is mostly influenced by governance and implementation capacity [47].

## 4.6 Strength and limitations

This review is the first scoping review to summarise and provide an overview of the factors associated with infant mortality, including other dimensions such as neonatal, perinatal and postneonatal mortality, within the Nigerian context. Secondly, no limit was set for publication date and study design, reducing selection bias. Also, using a socioecological model to report the factors affecting infant mortality at multiple social levels offers an understanding of the complexity of infant deaths and highlights how each level is facilitated and evaluated. Despite these strengths, this review did not report on medical causes and genetic risk factors because the model only allows for categorisations based on social and environmental inter-relationships. Another limitation of the study is its exclusive inclusion of articles published solely in English. Lastly, this review did not make use of a quality assessment tool, and as such, did not critically appraise the articles included, however, this also supports the review's objective, which is to provide an overview of existing and current studies on infant mortality in Nigeria.

## 5. Recommendation and conclusions

This review highlights areas that require attention and further research for researchers, policymakers, and other stakeholders in the health sector. From the review, it was observed that very few studies examined public policy-level factors while the majority focused on individual-level factors, resulting in several gaps in knowledge areas. The gaps identified from the review include the effect of maternal outcome after delivery on infant death, the role of household headship on infant mortality, especially for families in Nigeria that have women as breadwinners, the provision of 24-hour health services specifically for women who deliver at night time, adequacy of neonatal transport vehicles and referral services, the influence of electricity as a risk factor, the role of conflict and oil spills especially in the south-south region, and lastly, the effect of private and public health expenditure on infant mortality in Nigeria. Therefore, the implication of this scoping review for researchers is to pursue further research and assess the impact of the gaps mentioned above on infant mortality in Nigeria. Additionally, a future systematic review is recommended to empirically appraise the effectiveness of all child health interventions and policies implemented in Nigeria to determine the strategies that contribute to the reduction of infant mortality in the country.

On the other hand, this review offers insights for policymakers by suggesting areas within each level of the SEM that require improvement. For example, policies that address access to protective healthcare services should be revised and improved to benefit both mother and

child. Secondly, there is a need for the Nigerian government to increase the public health budget and expenditure as this will help alleviate the burden of out-of-pocket spending on citizens. Lastly, the efforts of the Ministry of Health should be directed at developing evidence-based interventions that will promote child well-being and facilitate the achievement of the SDG's goal 3.2 by 2030.

The results presented in this review suggest that several factors influence infant mortality at different socioecological levels in Nigeria. Several geographical and socioeconomic disparities put specific populations at risk of infant mortality than others, and these disparities are observed at the individual, interpersonal, organisational, community and political levels. This review, therefore, suggests that infant mortality can be reduced if health measures and interventions are designed to impact socioeconomic inequalities.

## Supporting information

**S1 Table. Search strategy and search histories applied to databases.**
(DOCX)

**S1 Appendix. Scoping review protocol.**
(DOCX)

**S2 Appendix. PRISMA checklist.**
(DOCX)

## Acknowledgments

The authors would like to acknowledge the MSc in Public Health Department in the Faculty of Education and Health Sciences at the University of Limerick.

## Author Contributions

**Conceptualization:** Loveth Dumebi Nwanze, Nuha Ibrahim.

**Investigation:** Loveth Dumebi Nwanze.

**Methodology:** Alaa Siuliman.

**Supervision:** Nuha Ibrahim.

**Writing – review & editing:** Loveth Dumebi Nwanze, Nuha Ibrahim.

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
