## [Decision Letter · Decision Letter 0]

25 Jul 2023

PONE-D-23-05730Factors affecting infant mortality in Nigeria: A scoping review.PLOS ONE

Dear Dr. Nwanze

Thank you for submitting your manuscript to PLOS ONE. After careful consideration, we feel that it has merit but does not fully meet PLOS ONE’s publication criteria as it currently stands. Therefore, we invite you to submit a revised version of the manuscript that addresses the points raised during the review process.

We look forward to receiving your revised manuscript.

Kind regards,

Kahsu Gebrekirstos Gebrekidan

Academic Editor

PLOS ONE

Journal Requirements:

Additional Editor Comments (if provided):

Please follow the submission guideline of Plose ONE when you write the subtopics, revise materials and methods, results, conclusion

The referencing needs revision, some looks incomplete and some are very old references. Therefore, the manuscript will be benefited with with revision of all references.

Reviewers' comments:

Reviewer's Responses to Questions

**Comments to the Author**

1. Is the manuscript technically sound, and do the data support the conclusions?

Reviewer #1: Partly

Reviewer #2: Yes

2. Has the statistical analysis been performed appropriately and rigorously? 

Reviewer #1: N/A

Reviewer #2: Yes

3. Have the authors made all data underlying the findings in their manuscript fully available?

Reviewer #1: Yes

Reviewer #2: Yes

4. Is the manuscript presented in an intelligible fashion and written in standard English?

Reviewer #1: No

Reviewer #2: Yes

5. Review Comments to the Author

Reviewer #1: General- The subject matter is important and the relevance has been demonstrated by the authors in the introduction segment. The authors however need to do some more searches of some databases specific for African literature. The authors have mixed UK and American English spelling, and some sentences are rather harsh and need to be rewritten.

Some segments are not clear in the Introduction segment and the authors attention has been drawn to these. Also some studies were excluded based on language. It is suggested the authors translate as these studies may be relevant.

The Data Extraction Chart showing studies are arranged in no order and it is suggested that authors arrange by year.

In the result segment, the figures have no titles, and there is a large chunk of prose repetition of Table content (Table 3),

For the Discussion, the authors did not compare the factors identified in some instances with other studies nor proffer reasons for these findings (birth outcomes, immunization). Also the author should try and relate these factors to the Nigerian statistics. The paragraphs are too long and the authors often discuss more than one factor in a paragraph making it hard to read.

References: The authors need to address errors in titles like all capital letters (4,22, 63) or first letters of words as capitals (17, 22,23, 27, 32,46, 50,52,53, 62,64,71. 74,81, 85, 86, 88, 92, 93, 96 and, repeats of author name (25) or missing journal name (64)

Reviewer #2: This manuscript title "Factors affecting infant mortality in Nigeria: A scoping review" is well written and rigorously conceived. However, there are syntax issues to be recognized and corrected.

Title: The title require minor but very important modifications representing correct syntax for such a study and appropriate alignment with the study design. Suggested modification is "Factors associated with infant mortality in Nigeria: A scoping review" Basis for this is all the studies included were observational studies with one quasi-experimental design. The outcomes from these studies can only infer association and not causation. Therefore a befitting title must reflect the syntax of association. Kindly reflect this in all places the word "affect" or "affecting" is used in the body of the manuscript.

Abstract: The abstract should be structure as: Background, Objective, Methods and Findings.

Editorial corrections are recommended in the abstract. For example, to maintain consistency in methodological syntax, the second and third sentences should read thus; "The risk of an infant dying before attaining one year of age is estimated to be six times higher in Africa than in Europe. Nigeria recorded an infant mortality rate of 72.2 deaths per 1,000 live births in 2020 in contrast with global estimate of 27.4 per 1,000 live births".

The next statement related to objective of the study would benefit from a modification to read thus: "This scoping review sought to identify and summarise the breadth of evidence available on factors associated with infant mortality in Nigeria".

The last sentence of the abstract should read: "Factors related to the individual, interpersonal, organisational, community and public policy levels were associated with infant mortality in Nigeria." Please note that only experimental study designs can unequivocally infer cause-effect relationship. Therefore such review, if it were focused exclusively on experimental studies such as randomized controlled trials or quasi-experimental design, then findings from such studies would be measuring and concluding on effects or influences of the treatments applied.

Introduction: The second sentence should read: "The number of infants that die before attaining the age of one year for every 1,000 live births in a year is known as Infant Mortality Rate (IMR) (2)."

The use of abbreviation PCC in section 2.1 Defining the research question should have been written fully. This should be "The Population-Concept-Context (PCC) framework applied defines the scope as infants mortality in the context of Nigeria".

3.3 Synthesis of Results: Please be mindful of the use of the word "affecting" as used throughout the manuscript, the review can only demonstrate that in the presence of certain antecedent factors are similarly associated with certain outcomes but cannot establish these to be the cause of the outcome. This is the limitation imposed by the design of the review. If the review was on experimental studies exclusively, then the appropriate syntax would apply.

General observations: While the socioecological model guided the review, it had only a contextual role to play but did not provide more technical theoretical framework to further define, for instance; intrapersonal level of personal-level disposition of health behaviour or health seeking behaviour of which the health belief model or information-motivation-behavioural skills would have contributed significantly in elucidating the dynamics at that level. At the interpersonal level, would have benefitted from relationships providing support or otherwise by including the Theory of planned behaviour. At the community and systems level would have also benefited by considering reinforcing and enabling factor offered by the Predisposing, reinforcing, enabling constructs in educational Diagnosis and Evaluation offered the Green et al., 1980/2008 ecological model.

Carefully read through the manuscript for language editorial cleaning. Well done.

6. PLOS authors have the option to publish the peer review history of their article (what does this mean?). If published, this will include your full peer review and any attached files.

Reviewer #1: **Yes: **Rasheedat Ibraheem

Reviewer #2: **Yes: **Nnodimele Onuigbo ATULOMAH

---

## [Author Response · Author response to Decision Letter 0]

8 Sep 2023

We would like to express our sincere appreciation to you and the esteemed reviewers for dedicating your valuable time to review our manuscript. Your constructive feedback has been instrumental in improving the quality of our work. We have carefully considered each of the points raised and have made the following responses and revisions:

1. Consistency in decimal usage: We appreciate the suggestion for consistency in decimal usage. We have revised the manuscript to maintain a single decimal format throughout the text.

2. Role of population: We have removed the reference to the population, and the sentence now reads, " Despite being one of the largest economies in Africa, with an estimated GDP of $448 billion in 2019, the socioeconomic indicators of Nigeria remain low, especially child mortality (10)."

3. Reference to several studies: In the sentence, "Several studies have traced the high mortality rates of infants in Nigeria to demographic and socio-economic factors, the most common being poverty/socioeconomic status and poor health service delivery," we have revised it to specify the two studies that were referenced. It now reads, " Studies have traced the high mortality rates of infants to demographic and socio-economic factors, the most common being poverty/socioeconomic status and poor health service delivery (11, 12)."

4. Attribution to Nigeria: We understand the concern about attributing factors to Nigeria in the introduction. We have rephrased the relevant portions to focus on the general factors associated with high infant mortality rates without singling out Nigeria.

5. Avoiding personalizing pronouns: We have rephrased the sentence as follows: " To estimate the burden of societal factors on neonatal deaths, Upadhyay et al. adopted the three-delay model in their study. The findings indicated that a delay in seeking care (level 1 delay) and a delay in receiving quality treatment at a health facility (level 3 delay) were significant contributors to neonatal mortality compared to a delay in reaching a health facility (level 2 delay) (14)"

6. Expanding search sources: We appreciate the recommendation to broaden our search sources. However, the authors had previously searched African Journal online (AJOL) during the initial study identification but did not utilise the database as there was no provision for advanced search to utilise the proposed search strategy. Also, we did not see the need to include the other mentioned databases as ones used for the study provided a comprehensive literature review and scope.

7. Translation of non-English studies: The inclusion criteria set by the authors was to exclude articles written in any language other than English as this is the official language used in Nigeria. Therefore, to include non-English studies would go against the initial inclusion criteria set.

8. Reordering data extraction chart: The data extraction chart has been reordered based on publication year, with the oldest studies appearing first.

9. Appropriate figure labelling: We have ensured that all figures are appropriately labelled. According to the submission guidelines, figures are not to be added in the main manuscript, but rather a figure caption must be inserted in the text of the manuscript. The titles seen after the references are for supplementary information and not for the figures, which have been attached separately.

10. Avoiding repetition: We have rephrased the prose on pages 16 to 18, presenting the information in the results section with appropriate references to Table 3.

11. Comparisons and reasons: We have enhanced and expanded the discussion section by comparing our findings with previous studies and elaborating on each factor identified.

12. Relating factors to Nigerian statistics: We have incorporated relevant Nigerian statistics into the discussion of identified factors, to provide a more comprehensive context for our findings.

13. Paragraph organization: We have reorganized the discussion section, breaking down lengthy paragraphs and ensuring that each paragraph focuses on a single factor, enhancing readability.

14. Addressing errors in titles: We have meticulously reviewed and rectified errors in titles, including capitalization and formatting issues, to conform to the journal's guidelines and standards.

We believe that these revisions address the concerns raised by the reviewers and significantly enhance the overall quality and clarity of our manuscript. We sincerely appreciate your guidance and feedback, which have been instrumental in refining our work. We look forward to your further guidance during the subsequent review process and hope that our revised manuscript will meet the high standards of your esteemed journal.

Thank you once again for your valuable input and consideration.

Sincerely,

Loveth Nwanze

---

## [Decision Letter · Decision Letter 1]

26 Sep 2023

PONE-D-23-05730R1Factors associated with infant mortality in Nigeria: A scoping review.PLOS ONE

Dear Dr. Lovelth,

Thank you for submitting your manuscript to PLOS ONE. After careful consideration, we feel that it has merit but does not fully meet PLOS ONE’s publication criteria as it currently stands. Therefore, we invite you to submit a revised version of the manuscript that addresses the points raised during the review process.

We look forward to receiving your revised manuscript.

Kind regards,

Kahsu Gebrekidan

Academic Editor

PLOS ONE

Journal Requirements:

Reviewers' comments:

Reviewer's Responses to Questions

**Comments to the Author**

1. If the authors have adequately addressed your comments raised in a previous round of review and you feel that this manuscript is now acceptable for publication, you may indicate that here to bypass the “Comments to the Author” section, enter your conflict of interest statement in the “Confidential to Editor” section, and submit your "Accept" recommendation.

Reviewer #1: (No Response)

Reviewer #2: All comments have been addressed

2. Is the manuscript technically sound, and do the data support the conclusions?

Reviewer #1: Yes

Reviewer #2: Yes

3. Has the statistical analysis been performed appropriately and rigorously? 

Reviewer #1: Yes

Reviewer #2: Yes

4. Have the authors made all data underlying the findings in their manuscript fully available?

Reviewer #1: Yes

Reviewer #2: Yes

5. Is the manuscript presented in an intelligible fashion and written in standard English?

Reviewer #1: Yes

Reviewer #2: Yes

6. Review Comments to the Author

Reviewer #1: (No Response)

Reviewer #2: All remarks and suggestions have been addressed by the authors. Congratulation for meticulously addressing all my concerns to my satisfaction.

7. PLOS authors have the option to publish the peer review history of their article (what does this mean?). If published, this will include your full peer review and any attached files.

Reviewer #1: No

Reviewer #2: **Yes: **Nnodimele Onuigbo ATULOMAH

---

## [Author Response · Author response to Decision Letter 1]

18 Oct 2023

Loveth Nwanze

University of Limerick

Ireland.

12 October 2023

The Academic Editor,

PLOS ONE

Dear Kahsu Gebrekidan,

We would like to extend our heartfelt appreciation to both you and the esteemed reviewers for generously dedicating your valuable time to reviewing our manuscript and providing invaluable feedback. Your meticulous evaluation has played a pivotal role in enhancing our submitted paper's quality and clarity, particularly, the final comments helped improve the methodology and limitation sections.

We have taken great care to thoroughly address each of the points raised, and we have documented our responses and revisions in the table below, with the reviewers' feedback indicated in black and our corresponding author responses highlighted in blue. These revisions have been strategically made to align with the reviewers' concerns, ultimately significantly improving the overall quality and clarity of our methodology and limitations sections.

We eagerly anticipate your continued guidance throughout the subsequent stages of the review process, and we are committed to ensuring that our revised manuscript upholds the rigorous standards of your esteemed journal.

Once again, we extend our gratitude for your invaluable input and consideration.

Kind Regards,

Loveth Nwanze

---

## [Decision Letter · Decision Letter 2]

2 Nov 2023

Factors associated with infant mortality in Nigeria: A scoping review.

PONE-D-23-05730R2

Dear Dr. Loveth,

We’re pleased to inform you that your manuscript has been judged scientifically suitable for publication and will be formally accepted for publication once it meets all outstanding technical requirements.

Kind regards,

Kahsu Gebrekidan

Academic Editor

PLOS ONE

Additional Editor Comments (optional):

Reviewers' comments:

Reviewer's Responses to Questions

**Comments to the Author**

1. If the authors have adequately addressed your comments raised in a previous round of review and you feel that this manuscript is now acceptable for publication, you may indicate that here to bypass the “Comments to the Author” section, enter your conflict of interest statement in the “Confidential to Editor” section, and submit your "Accept" recommendation.

Reviewer #1: All comments have been addressed

2. Is the manuscript technically sound, and do the data support the conclusions?

Reviewer #1: Yes

3. Has the statistical analysis been performed appropriately and rigorously? 

Reviewer #1: (No Response)

4. Have the authors made all data underlying the findings in their manuscript fully available?

Reviewer #1: Yes

5. Is the manuscript presented in an intelligible fashion and written in standard English?

Reviewer #1: Yes

6. Review Comments to the Author

Reviewer #1: The authors have addressed the concerns earlier raised. The manuscript had good flow and clarity.

A tiny edit-

1- The authors use these interchangeably (lower-middle-income countries and low-middle-income countries). Kindly stick to one (Check abstract and introduction)

7. PLOS authors have the option to publish the peer review history of their article (what does this mean?). If published, this will include your full peer review and any attached files.

Reviewer #1: No

---

## [Editor Report · Acceptance letter]

6 Nov 2023

PONE-D-23-05730R2 

Factors associated with infant mortality in Nigeria: A scoping review. 

Dear Dr. Nwanze:

I'm pleased to inform you that your manuscript has been deemed suitable for publication in PLOS ONE. Congratulations! Your manuscript is now with our production department. 

Kind regards, 

on behalf of

Dr. Kahsu Gebrekidan 

Academic Editor

PLOS ONE